# CONNECTING REPRESENTATION AND GENERATION VIA MASKED VISION-LANGUAGE TRANSFORMER

## ABSTRACT

Recently, there has been great progress in the self-supervised pre-training of multimodal representation models that understand image and language jointly. One particularly popular application of such models is text-to-image generation, which is typically obtained via a two-stage process: in the first stage, a representation model is trained via self-supervised objectives; then in the second stage, a conditional generative decoder is trained on top of the representation to generate natural images. In this work, we aim at bringing representation learning and conditional generation together by unifying the two stages into a single model and training objective. We present UPGen, a unified pre-trained model for both representation learning and generation. UPGen is trained with a simple masked token prediction objective on a flexible mixture of image and language data. We use a pre-trained VQGAN image tokenizer to convert images into discrete tokens, then train a masked token prediction model on both paired image-text datasets and unpaired language datasets, using randomly sampled mask ratios. We show that this masked token prediction model can be directly used to generate images and language by iteratively re-masking and predicting the masked tokens. We demonstrate empirically that UPGen serves as both a good representation learning model and a generative model for both image and language.

## 1 INTRODUCTION

With the rapid improvement of deep learning architecture and accelerator hardware, researchers have made significant progress in self-supervised representation learning from image and text data (Radford et al., 2021; Geng et al., 2022; Mu et al., 2021; Wang et al., 2022). Such models are trained to jointly understand language and image data and learn generalizable representations that transfer across image and language modalities, and thus can be applied to a wide variety of downstream tasks. One interesting task that has gained popularity recently is text-to-image generation, where the model is given a text prompt and generates an image that corresponds to the text prompt's description (Saharia et al., 2022; Ramesh et al., 2021; Yu et al., 2022). This task is particularly attractive because it enables a human to directly interact with the model and inspect its understanding of language and image, providing great tools for artistic creations.

Driven by the successes of representation learning, text-to-image models have achieved impressive progress (see *e.g.*, Saharia et al., 2022; Ramesh et al., 2021; Yu et al., 2022). However, the drawbacks of these text-to-images models are due to the pipeline having two stages. In the first stage, a representation model is trained with self-supervised pre-training objectives. In the second stage, a diffusion (Saharia et al., 2022; Ramesh et al., 2021) or autoregressive (Yu et al., 2022) generative model is trained conditioned on top of the (typically frozen) pre-trained representation. Such a two-stage training pipeline requires more hyperparameters to be tuned, and introduces extra complexity in developing models.

To address the above limitations, we propose UPGen, a simple but effective framework that unifies pre-training for representation and generation. Using a pre-trained VQGAN model (Esser et al., 2021), we convert an image into a sequence of discrete tokens and concatenate the image tokens and language tokens into a single sequence. We then train an encoder-only transformer model using a simple masked token prediction objective on the concatenated sequence. The masked token prediction objectives produces good representations for downstream tasks. Furthermore, we

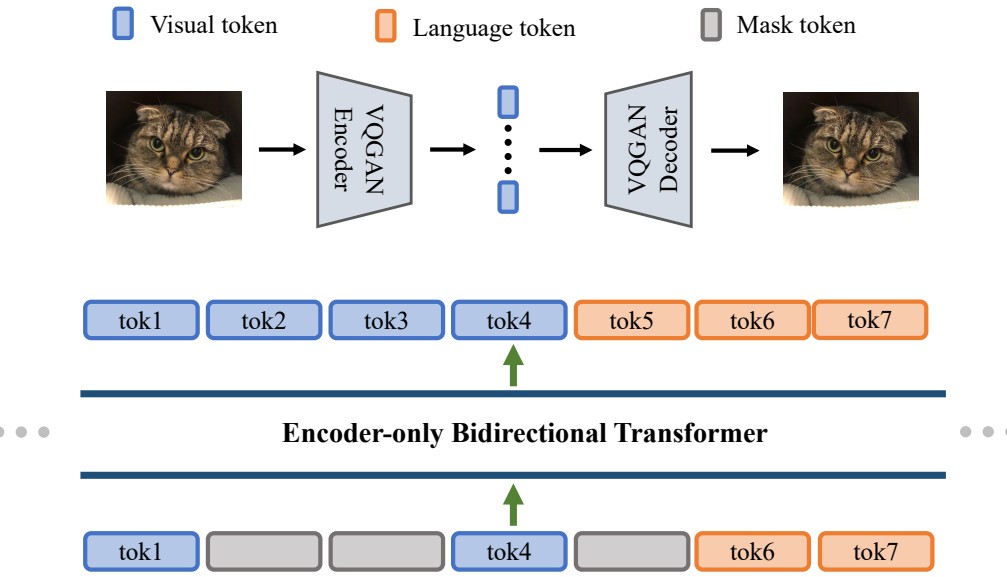

Figure 1: UPGen consists of a pre-trained VQGAN tokenizer that converts images into discrete tokens and an encoder-only transformer that processes the image tokens and language tokens jointly. The concatenated image and language sequence are randomly masked according to a uniformly sampled ratio, and the transformer is trained to predict the masked tokens.

show that UPGen can be directly used for conditional and unconditional generation of images and language without having to train extra components. To achieve this, we apply an iterative refinement strategy by repeatedly re-masking and regenerating the masked tokens, following the approach of MaskGIT (Chang et al., 2022).

In this work, we provide a large-scale empirical study of UPGen on a mixture of paired image-text datasets and unpaired language datasets. We find that UPGen learns generalizable representations that transfer to a wide variety of tasks, including image classification, text-guided image inpainting, and image-to-text generation. We also demonstrate that UPGen can generate high-quality images with and without language conditioning on language prompts. While achieving competitive results, UPGen does not perform as well as state-of-the-art methods on each of the downstream tasks due to being trained on much smaller datasets. However, to the best of our knowledge, UPGen is the first model that combines representation learning, image-to-text generation, text-to-image-generation, and unconditioned image-generation into a single model and training objective. Scaling UPGen to larger datasets and model size is left as promising future work.

## 2 RELATED WORKS

**Self-supervised learning via masked modeling.** Ever since the introduction of Transformers (Vaswani et al., 2017), self-supervised pre-training has made significant progress in the recent years. A particularly popular style of self-supervised learning is masked modeling, where the input example is partially masked and the model is trained to predict the masked part from the unmasked part. Masked modeling first sees its success in natural language processing (NLP), with large language models like BERT (Devlin et al., 2018), RoBERTa (Liu et al., 2019), T5 (Raffel et al., 2020) and UL2 (Tay et al., 2022) that learns highly generalizable representations that transfer well to various of downstream tasks. Inspired by the effectiveness of these NLP models, researchers have taken the masked modeling ideas into computer vision and also met with great success. Vision transformers (Dosovitskiy et al., 2020) introduces the transformer architecture into computer vision and make the natural analogy between language tokens and patches of natural images. Building on this work, BEiT (Bao et al., 2021) proposes to apply the masked token prediction on vision transformers, taking image patches as input and predicting the output of a discrete tokens produced by a VQVAE (Van Den Oord et al., 2017) model. Recently, MAE (He et al., 2022) further removes the need of a discrete image tokenizer by directly predicting the patch pixel output using an encoder-decoder architecture. Our UPGen first tokenizes the image into discrete tokens by using a pre-trained VQGAN (Esser et al.,

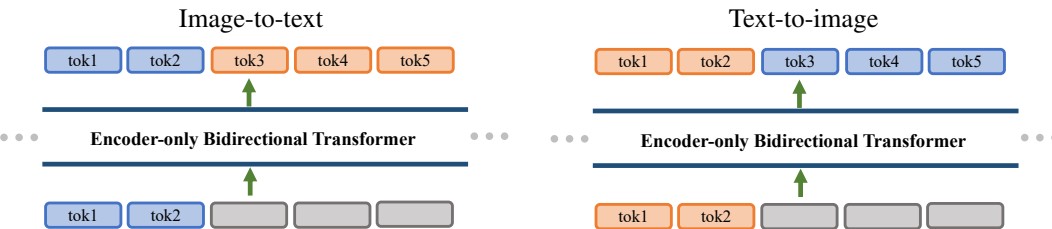

Figure 2: Pre-trained UPGen can be directly used for cross-modality generation: Image-to-text generation (**left**) and text-to-image generation (**right**) by inputting a sequence of masked tokens.

2021) model, and then follows the simple masked token prediction objective of BERT. UPGen differs from BEiT mainly in the way that it uses the discretized image tokens as both input and output, which facilitates the generation process. In addition, our model is trained with both image and language data, which connects the information from different modalities together.

**Multimodal representation learning for language and image.** While self-supervised learning on image or language data alone has proven great success in extracting useful information and producing generalizable representations, recent works have shown that the such capability can be greatly improved if we connect these two modalities modalities together. Multimodal self-supervised learning models achieves this by jointly learning on paired data of two modalities. One popular approach for multimodal pre-training is contrastive learning. Contrastive learning approaches such as CLIP (Radford et al., 2021), SLIP (Mu et al., 2021) and FLAVA (Singh et al., 2021) connects representation between different modalities together by maximizing the similarity between positive example pairs and minimizing the similarities between negative pairs. Such objectives are limited to paired image-text datasets, and extending the pre-training to unpaired data often requires the introduction of additional objectives. Recently, with the advancement of masked modeling techniques for images, masked modeling has gained a lot of popularity as an alternative objective for contrastive learning due to its simplicity and flexibility. M3AE (Geng et al., 2022) extends the MAE approach to both image and language by combining image patches and language tokens as a single sequence. BEiTv3 (Wang et al., 2022) also combines image patches and language tokens, but predict discretized image tokens from a VQVAE model as output. Both of these models uses simple masked token prediction objective. Unlike M3AE and BEiTv3, UPGen uses discretized image tokens for both model input and output in order to perform generation. Moreover, UPGen is trained with uniformly sampled mask ratios from 0 to 1, removing the need to tune mask ratio for each modality.

**Text-to-image generation.** Text-to-image generation has been a popular field with various approaches being proposed recently. Typically, text-to-image models follow the general pattern of pre-training a language encoder model via self-supervised objective and then training a generative image decoder conditioned on the language representation. Therefore, these models can be characterized by their language encoder type and their generative decoder type. From the language encoder perspective, DALL-E (Ramesh et al., 2021), DALL-E 2 (Ramesh et al., 2022) and Stable Diffusion (Rombach et al., 2021) uses the multimodal CLIP as their language encoder. Imagen (Saharia et al., 2022) uses T5 (Raffel et al., 2020), a pure language model, and Parti (Yu et al., 2022) trains a BERT-like transformer as its language encoder. GLIDE (Nichol et al., 2021) trains a transformer with the generative model in an end-to-end way to process language. From the generative model perspective, most approaches including DALL-E 2, GLIDE, Stable Diffusion and Imagen uses diffusion models (Ho et al., 2020) as its generative decoders. DALL-E and Parti uses autoregressive models in the discretized image token space as the generative decoders. Unlike these prior approaches, our model combines the representation encoder and generative decoder into a single transformer model. We show that the UPGen can complete both representation learning and generative tasks by training on a single objective of masked token prediction.

## 3 UNIFYING REPRESENTATION PRE-TRAINING AND GENERATION

In this section we introduce our method UPGen, which is an encoder-only transformer model that takes the masked sequence of image and language tokens as input and predicts the masked tokens. We summarize the architecture and training objectives in Figure 1.

**VQGAN-based image tokenization.** In order to apply the transformers architectures to images which do not exhibit a natural sequence structure, prior works typically use one of two approaches: slice the image into patches and treat these patches as a sequence, or use a pre-trained discrete representation model such as VQVAE (Van Den Oord et al., 2017), DVAE (Ramesh et al., 2021) or VQGAN (Esser et al., 2021) to convert the image into a sequence of discrete tokens. In this work, we choose the latter approach for many practical reasons. Firstly, the discrete representations bring the images closer to language in terms of format, allowing us to use the same input processing and cross-entropy loss function for both image tokens and language tokens. Moreover, the discrete-token classification output of our model naturally forms a probability distribution with high degree of freedom, allowing us to directly sample from the predicted token classes during generation without having to rely on extra probability distribution assumptions. For simplicity, we directly use the open-source pre-trained model from MaskGIT (Chang et al., 2022), which is a highly effective ConvNet-based VQGAN model trained on the ImageNet 1K dataset (Deng et al., 2009). We keep the VQGAN model fixed throughout our training process.

**UPGen Transformer architecture.** After using the pre-trained VQGAN to convert an image into a sequence of tokens, we concatenate the image tokens and language tokens into a single sequence. For unpaired examples, we simply take the sequence of image or language tokens. We apply the standard trainable word embedding layer to obtain a sequence of vectors. In addition to the standard positional embeddings, we also add modality-type embeddings to enable the model to distinguish between two modalities, following the approach of M3AE (Geng et al., 2022). We feed the sequence of vectors into an encoder-only Transformer network, and apply a final linear projection on the output of the Transformer to obtain the classification logits.

**Masked token prediction training.** We train UPGen with a simple masked token prediction objective. With the combined sequence of image and language tokens, we first randomly sample a mask ratio uniformly between 0 and 1 for each example in the batch, following the approach of MaskGIT (Chang et al., 2022), and then randomly replace the tokens in the combined sequence with a special mask token according to the ratio. We feed the masked sequence into the Transformer, and supervise the output with the ground-truth token only at the masked position, similar to BERT (Devlin et al., 2018).

Specifically, let $\mathbf{X} = [x_i]_{i=1}^N$ denote the latent tokens obtained by processing the image with the VQGAN encoder, where $N$ is the length of the reshaped token matrix, and $\mathbf{Y} = [y_i]_{i=1}^K$ denote the text tokens obtained via the text tokenizer, where $K$ is the number of text tokens. During training, we sample a subset of tokens and replace them with a special [MASK] token. Let $\mathbf{M} = \{0,1\}^{(N+K)}$ denote a randomly generated mask, we mask the token $x_i$ or $y_j$ if $\mathbf{M}[i] = 1$ or $\mathbf{M}[j] = 1$. Let $\mathbf{Z} = [\mathbf{X}, \mathbf{Y}]$ denote the sequence of image tokens and text tokens, and $\hat{\mathbf{Z}} = \mathbf{Z} \odot \mathbf{M}$ denote the sequence of unmasked tokens. The training objective is to minimize the negative log-likelihood of the masked tokens:

$$\mathcal{L} = -\mathop{\mathbb{E}}_{\mathbf{Z} \in \mathcal{D}} \Big[ \sum_{\forall i \in [1, N+K], \mathbf{M}[i]=1} \log p(z_i | \hat{\mathbf{Z}}) \Big], \tag{1}$$

where $\mathbf{Z} \in \mathcal{D}$ denotes over all image-text sequences. Note that when $\mathbf{Z} = \mathbf{X}$, Equation 1 reduces to MaskGIT (Chang et al., 2022) which does image-only masked token prediction. When $\mathbf{Z} = \mathbf{Y}$, Equation 1 reduces to training a BERT (Devlin et al., 2018) on language tokens with random mask ratio. Since the model is agnostic to what $\mathbf{Z}$ consists of, *i.e.*, it can be image-text pair, text-only and image-only data, we train UPGen on not only image-text pairs but also text-only data to enable better language understanding.

**Iterative cross-modality generation.** During inference, since the model has been trained to model any set of unconditional and conditional probabilities, we can sample any subset of tokens per sampling iteration, from the extreme cases of sampling all tokens (independent) versus sampling each token at a time (autoregressive). We follow MaskGIT (Chang et al., 2022) to use a confidence-based sampling mechanism. Specifically, at each iteration, we only keep the predictions that the model is most confident with and discard the rest of predicted tokens and replace them with mask. We repeat this process until all tokens are generated.

We can control the model to condition on image tokens to predict text tokens by inputting a sequence of image tokens followed by a sequence of [MASK] tokens. The same can be done for text-to-image generation. The process is illustrated in Figure 2. For text-to-image generation, we visualize the tokens generated at intermediate iterations in Figure 3.

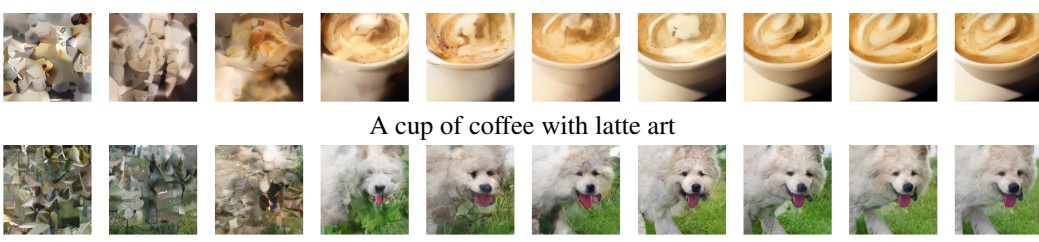

A cup of coffee with latte art

A large dog in the park

Figure 3: Visualization of the iterative text-to-image generation process of UPGen. The text prompt is shown below the image generation process.

## 4 EXPERIMENTS

In this section, we conduct an empirical study of UPGen and present the results. Since UPGen unifies representation learning and generation into a single model and training objective, it is important to evaluate its performance from both perspectives. Hence, in designing the experiments, we aim to answer the following questions: **(1)** As a representation learning model, does UPGen learn generalizable visual representations that transfer well to downstream tasks? **(2)** As a generative model, can UPGen generate high-quality image and text samples?

To answer these questions, we pre-train UPGen on a large and diverse dataset of image-language pairs. In addition to the paired data, we also use a combination of unpaired language-only data. We evaluate the representation quality of pre-trained UPGen models on image classification tasks, and we evaluate the image and text generation by inspecting the generated samples both qualitatively and quantitatively.

### 4.1 UPGEN PRE-TRAINING

**Datasets.** In pre-training UPGen, we use a combination of image-text paired datasets and text-only unpaired datasets. For the image-text paired dataset, we use the publicly available Conceptual Caption 12M (CC12M) (Changpinyo et al., 2021), Redcaps (Desai et al., 2021), and a 15M subset of the YFCC100M (Thomee et al., 2016) selected according to (Radford et al., 2021). For Conceptual Captions 12M, since the dataset is provided in the form of online image URLs, we are only able to retrieve 10M images due to some expired URLs. The combination of these three datasets give us a total of 36M image-text pairs.

For the unpaired text-only dataset, we follow the practice of BERT (Devlin et al., 2018) and use a mixture of multiple publicly available text corpus. Specifically, we combine the text datasets of Wikipedia (Foundation), CC-News (Hamborg et al., 2017), OpenWebText (Gokaslan & Cohen) and Toronto BookCorpus (Zhu et al., 2015).

**Model architecture and input processing.** UPGen's main model is a bi-directional encoder-only transformer (Vaswani et al., 2017) network. We adopt the convention of Vision Transformers (Dosovitskiy et al., 2020) and use a ViT-Large sized model. The pre-training takes the combined sequence of image tokens and language tokens as input after applying a fixed sinusoidal positional positional embedding and learnable type embedding. For the VQGAN image tokenizer, we use the pre-trained model from MaskGIT (Chang et al., 2022), which is a convnet-based encoder and decoder VQGAN pre-trained on ImageNet. The tokenizer converts a $256\times256$ image into 256 discrete tokens, where the codebook size is 1024. We list the detailed architecture and input processing information in Appendix A.

**Pre-training.** For all models, we pre-train for 50 epochs on the combined dataset with cross-entropy loss on the masked tokens. For paired image-text tokens, we sample a mask ratio uniformly between 0 and 1 every time and for unpaired text tokens we use a fixed mask ratio of 15%. For the default model, we apply a weighting of 1.0 for the loss on the image tokens, a weighting of 0.1 on the paired text tokens and a weighting of 0.01 on the unpaired text tokens. Detailed training hyperparameter information can be found in Appendix A.

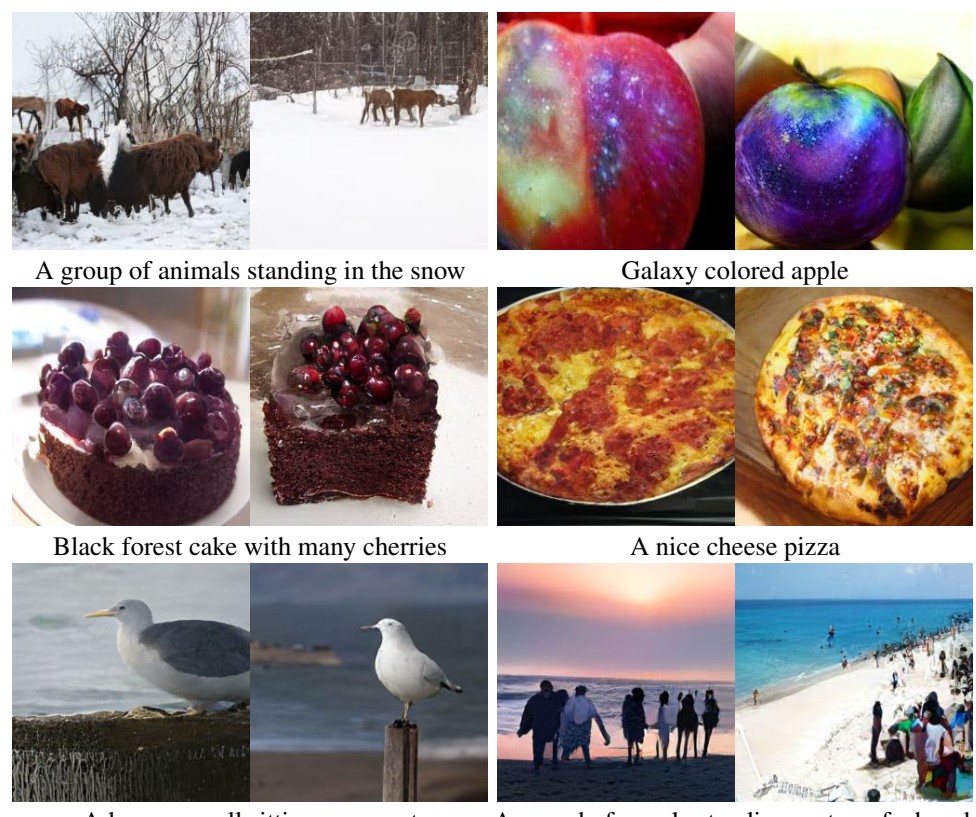

| A group of animals standing in the snow | Galaxy colored apple |
| Black forest cake with many cherries | A nice cheese pizza |
| A large seagull sitting on a post | A crowd of people standing on top of a beach |

Figure 4: Examples of text-to-image generation of UPGen. The two images for the each prompt are generated with different random seeds.

## 4.2 REPRESENTATION LEARNING

Following the convention in prior works, we evaluate the representation learning performance of our model by performing linear classification on top of the learned representation on ImageNet 1K. We freeze the weights of the UPGen pre-train and take the embeddings of all the intermediate layers. We concatenate these embeddings and train a linear classification layer on top. We summarize the results in Table 1. The results show that UPGen achieves comparable performance to prior works in representation learning, indicating that UPGen can serve as a simple but strong alternative to existing visual representation learning methods. We do note that CLIP outperforms our model significantly, and hypothesize that this is the result of CLIP being trained on a proprietary dataset 10 times larger than the one we use.

| Model | UPGen | UPGen (paired data only) | UPGen (image-only) |
|---|---|---|---|
| Accuracy | 70.18 | 70.18 | 70.17 |
| Model | CLIP (YFCC100M) | MAE (ImageNet) | M3AE (CC12M) |
| Accuracy | 83.9 | 73.5 | 64.1 |

Table 1: Linear classification accuracy on ImageNet 1K validation set. All models are based on ViT-L with the same number of parameters. In the top row, we present the accuracy of UPGen pre-trained with both paired and unpaired data, with unpaired data only and with image data only. In the second row, we show the performance of visual representation learning methods trained on other datasets. We see that UPGen reaches similar performance to other representation learning methods.

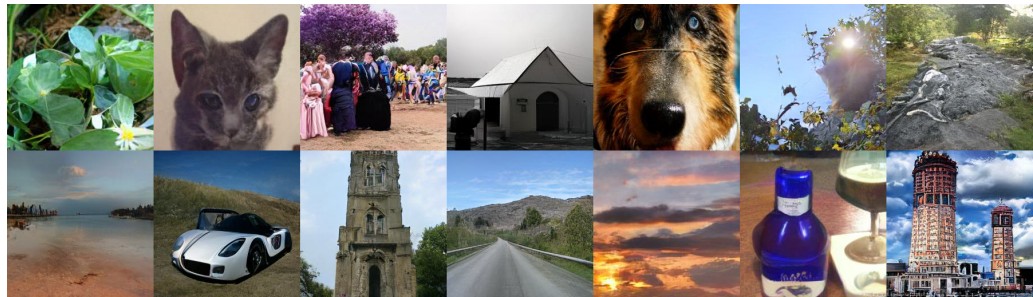

Figure 5: Examples of unconditioned image generation of UPGen. Even without conditioning on text prompts, UPGen is still able to generate images that contain diverse objects.

## 4.3    IMAGE GENERATION

**Qualitative results.** We perform qualitative evaluation on the image generation of UPGen. We present selected text-to-image generation results in Figure 4. We see UPGen is able to generate mostly coherent images that correspond well to the description of the text prompt. It can be observed that these images contain certain unnatural artifacts. We hypothesize that this is largely caused by the specific VQGAN image tokenizer we use, which is only trained on ImageNet and therefore is not able to encode objects not included in the ImageNet dataset. We also present unconditioned image generations in Figure 5. We see that even without conditioning on text prompts, UPGen can still generate images that contain natural objects. This further demonstrates the generality of our UPGen training objective.

**Zero-Shot FID score on MS-COCO** Following the convention in prior works of image-to-text generation, we use the FID (Heusel et al., 2017) score on MS-COCO (Lin et al., 2014) validation set as a quantitative metric of quality of generated images. To perform the evaluation, we sample 30K captions from the annotations of MS-COCO 2014 validation set and generate images from these captions. We then calculate the FID score against the original images in the validation set. This setting is known as zero-shot FID score since our model is not trained or finetuned on MS-COCO. We present the results in Table 2. While our model does not perform as well as prior methods for text-to-image generation, we note that our model is trained with a much smaller dataset, and to the best of our knowledge, UPGen is the first model that unifies representation learning and generation (of text and language, both conditioned and unconditioned) into a single model and training objective.

| Model | UPGen | DALL-E | StableDiffusion | GLIDE | DALL-E 2 | Imagen | Parti |
|---|---|---|---|---|---|---|---|
| FID-30K score | 65.25 | 28 | 12.63 | 12.24 | 10.39 | 7.27 | 7.23 |

Table 2: Zero shot FID-30K score evaluated on the MS-COCO validation set.

## 4.4    TEXT GENERATION

To evaluate the quality of image-to-text generation of UPGen, we take several images from the MS-COCO validation set and use our model to generate captions for these images. We perform the exact same sampling process as text-to-image generation, starting from the image tokens instead of text tokens as unmasked input. We present the results in Figure 6. While the text is not fluent as those generated by an autoregressive language model, we see that UPGen is able to capture important keywords that describe the content of the image.

## 4.5    ABLATION STUDIES

**Sampling temperature for image generation.** Following the practice of MaskGIT, we use a temperature to sample from the predicted logits during the generation process. This temperature hyperparameter has great influence on the quality of generated images. To find out the optimal

| Image prompt | Dataset annotation | Model output |
|---|---|---|
|  | the light display of the erupting volcano of stromboli. | this violent volcano going out of the atmosphere will just erupt! an active volcano - - called the volcano ' ' thundering lightning - has glow only been abandoned for two weeks. the human sense of sound with its active bodies lies within, if ever in the parts, or event. < person >, lightning wallpaper. |
|  | the who vintage yellow guitar - 0 | this vintage 1950s electric guitar is available to elements from unique parts of the body. we will welcome this to your life. we are all unique. made of diameter. 3's 6. 5 ' ', in gold top, in brass and engraved hardware, with custom hardware. electric guitar, guitar amp, guitar amp |
|  | tower grove park : beautiful landscaping throughout the park. | according to < person >, the houston zoo is one of the most beautiful zoos of america, but the zoo does. if it cannot afford for any terms of the budget, it will cost you too much money. if it is going to be saving us, i would wonder if we can visit it. - - |
|  | north sea beach, the hague, netherlands | i was looking for a walk, but decided to conveniently cross - my - break when i received this lovely, brief breeze perfectly between my legs - either going for a stroll, or not to walk in it, whether you should walk on the beach or on the place. if you like me, then go out. |

Figure 6: Examples of image-to-text generation using our method. The images are sampled from the CC12M dataset. **Left**. The sampled image prompt from CC12M. **Middle**. Text annotation of the image from CC12M. **Right**. Model-generated text based on the image prompt.

temperature, we measure the FID score on MS-COCO validation set of generated images on different temperatures, and present the result in Figure 7. The result suggests an optimal temperature of 20. Qualitatively, we find that having too low temperature makes the generation process more greedy by only sampling the most probable token, which result in less diverse images. If the temperature is too high, the generation process becomes too chaotic and the generated images are less coherent.

**The effect of unpaired text data.** While the flexibility of UPGen enables the use of both paired and unpaired image and text data in training, it is important to understand the effects of using additional unpaired text data during pre-training. We conduct an ablation study comparing the performance of UPGen pre-trained with and without unpaired text data. We measure the visual representation quality of these two models via ImageNet linear classification accuracy, and test the quality of text-to-image generation with the FID score on MS-COCO validation set. We present the results in Table 3. We see

| Training data | With unpaired text | No unpaired text |
|---|---|---|
| MS-COCO zero-shot FID-30K score | 65.25 | 67.32 |
| ImageNet linear classification accuracy | 70.18 | 70.18 |

Table 3: Ablation study of the effects of using additional unpaired language data in pre-training UPGen. We see that while the use of unpaired text data does not improve visual representation quality, it improves the quality of text-to-image generation which requires more complex language understanding.

that the use of unpaired data does not improve the quality of visual representations. This result is expected since the ImageNet classification task does not require understand of complex logic and reasoning in natural language, and simply matching keywords is sufficient. On the other hand, we observe that the use of unpaired text data improves the FID score for text-to-image generation, which is a task that requires stronger language understanding.

**The effect of loss weights.** During the pre-training of UPGen, although we use cross-entropy loss for both predicted image tokens and predicted language tokens, we find that it is important to balance the weight of these two losses. Specifically, we discover that having a lower weight on the language loss improves the quality of both visual representation and image generation. Following the previous ablation study, we present the effect of the language loss weights in Table 4. We hypothesize that this is caused by the text token prediction being a much simpler problem than image token prediction. This can also be observed by the training accuracy of text and token predictions. During the final steps of training, the masked text token prediction accuracy converges to 37% while the masked image token prediction accuracy stay around 15%. Hence, it is important to put more weights on the more difficult image token prediction problem.

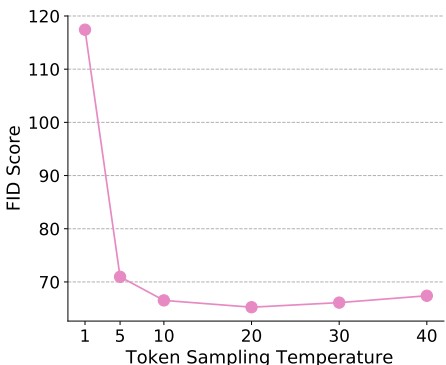

Figure 7: The effect of different sampling temperature on MS-COCO FID score.

| Text loss weight of UPGen | 1.0 | 0.1 |
|---|---|---|
| MS-COCO zero-Shot FID-30K score | 66.85 | 65.25 |
| ImageNet linear classification accuracy | 69.13 | 70.18 |

Table 4: Ablation study of the text loss weight during pre-training of UPGen. We see that using a smaller weight for language loss improves both visual representation and image generation.

## 5 CONCLUSION AND LIMITATIONS

In this work, we present UPGen, a simple masked Transformer-based method for multimodal tasks, including representation learning, image-to-text generation, and text-to-image generation. We conducted quantitative and qualitative experimental evaluations on UPGen and demonstrate competitive performance on a wide range of tasks. As UPGen is scalable and flexible, an interesting future work would be scaling up the amount of image-text, image-only and text-only data. A limitation of UPGen is that incorporating text-only data does not improve visual representation learning, even though training on additional language data should enable the model to better leverage cross-modal information. We believe resolving this would further improve the results of UPGen, and leave this as a future work.

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

## A  IMPLEMENTATIONS DETAILS

We provide our implementation and training details of UPGen in this section.

### A.1  IMPLEMENTATION

We implement our UPGen Transformers using Flax (Heek et al., 2020) and JAX (Bradbury et al., 2018). Our implementation is largely based on open source codebases, such as the released code for MaskGIT (Chang et al., 2022) and M3AE (Geng et al., 2022). For the VQGAN tokenizer, we directly use the open source ImageNet pre-trained VQGAN from MaskGIT for 256x256 image size.

### A.2  HYPERPARAMETERS

In this section we provide the training hyperparameters of UPGen. These configurations largely follow the training hyperparameters of MaskGIT (Chang et al., 2022).

| Hyperparameter | Value |
|---|---|
| Model size | ViT-Large |
| Dropout | 0.1 |
| Optimizer | AdamW |
| Base learning rate | 0.005 |
| Weight decay | 0.05 |
| Optimizer momentum | $\beta_1 = 0.9, \beta_2 = 0.95$ |
| Batch size | 4096 |
| Learning rate schedule | cosine decay |
| Epochs | 50 |
| Warmup epochs | 5 |
| Image data augmentation | Random crop and flip |
| Paired image loss weight | 1.0 |
| Paired text loss weight | 0.1 |
| Unpaired text loss weight | 0.01 |

Table 5: Hyperparameters for training UPGen.

### A.3  TRAINING AND COMPUTE RESOURCES

We train our model on Cloud TPUv3 pods on Google Cloud Platform through the TPU Research Cloud program. On a TPUv3-512 pod, UPGen using unpaired data takes 80 hours to train and UPGen with only paired data takes 45 hours to train.

