# OpenReview forum: "Connecting representation and generation via masked vision-language transformer"
_ICLR.cc/2023/Conference — Submitted to ICLR 2023_

### Official Review · Reviewer_DLC8 · 2022-10-20

**Confidence:** 4
**Clarity, Quality, Novelty And Reproducibility:** Fair
**Correctness:** 3
**Technical Novelty And Significance:** 2
**Empirical Novelty And Significance:** 3
**Recommendation:** 5

**Strength And Weaknesses:**

Strengths:

- This paper is the first one completing representation learning and generative tasks by training on a single objective of masked token prediction. This way of text-prompt image generation and image-prompt text generation is simple but elegant.

- This paper narrow the gap between generation and representation and has great potential in other image tokenizer and masked tokens prediction based model.

- The experiments are relatively rich and the proposed model can be applied to tasks like representation learning, conditional generation tasks between image and text and unconditional image generation simultaneously.

Weaknesses:

- There are some grammatical errors in this paper. For example: (1) typo in page 2, “Multimodal representation learning for language and image” paragraph, “if we connect these two modalities modalities together” -> “if we connect these two modalities together”; (2) in the second paragraph of the introduction, page 1, it says “is trained conditioned on top of”, the word “conditioned” should be modified; (3) “Following the convention of in prior works of image-to-text” -> “Following the convention of in prior works of text-to-image” (4) “and Zˆ = Z ⊙ M denote the sequence of unmasked tokens.”-> “and Zˆ = Z ⊙ M denote the sequence of masked tokens.””

- The authors skip the details of iterative strategy and image token decoding step, which seems just following MaskGIT and VQGAN. The only novelty in the model is the generation objective.

- What’s more, this work does not show the advantage over other generative model. The quality of generated images do not perform SOTA results. And the Linear Classification accuracy on ImageNet 1K does not achieve comparable performance to prior works. In comparison with the two stage, the authors only indicate that one stage models need less hyperparameters and has less complexity in developing models, which do not show the superiority of one stage models adequately. Although the authors strengthen that they train the model on a small dataset, they still need to compare with the prior works on the same dataset scale to prove their model’s performance and transferability.

- The authors point out UPGen learns generalizable representations that transfer to a wide variety of tasks in page 2. But in the paper, the authors only release the comparison of linear classification accuracy between prior works. More downstream task comparisons should be completed.

- The ablation studies should show more results of different loss weight of image and different mask ratio strategy. In this paper, the authors sample a mask ratio uniformly between 0 and 1 every time in paired image-text tokens and for unpaired text tokens we use a fixed mask ratio of 15%, so the expectation of the mask ratio in paired tokens is 50%. The author need to show the mask strategy of uniformly sampling is better than a fixed one. Also, this work applies a weighting of 1.0 for the loss on the image tokens, a weighting of 0.1 on the paired text tokens and a weighting of 0.01 on the unpaired text tokens, but in the ablation study they only show the accuracy of text loss weight in 1.0 and 0.1. So more weight combination should be compared in the study.

- From the experimental results in Table 1, the reviews can see that the performance of the proposed model is inferior to MAE under the similar data size. The cause of the phenomenon is lacked.

- The model does not perform as well as prior methods in the tasks it aimed to address. In the image to text task, the model can only capture important keywords and the generated text is not fluent with meaningless items in. Why?

-  Lack of experiments trained with image only data. For text only data, the additional unpaired data did not improve the representation learning performance as expected

Suggestions:

- The experiments results of the proposed model should be in bold to make it more clear for readers to follow.

- There are many pictures in the main text, such as figure 5: Examples of unconditioned image generation of UPGen. I am wondering if these pictures are necessary. In my opinion, the figures should be the something like tend chart and so on.

- The ablation studies only introduce  a smaller weight for language loss,and failed to consider using a smaller weight for image loss. It only proves the influence of text on image, not the influence of image on text.

- The model it trained on much smaller datasets,  and thus it can't perform as well as the SOTA methods on each downstream tasks. It is recommended to train with same dataset and compare their performance.

- The authors say the proposed model can reduce the number of hyperparameters need to be tuned and reduce the complexity of the development process of model. I recommend that the authors may increase the conviction of the experiments by experimenting the proposed model and other text-to-image models on the same datasets and comparing the speed and efficiency.

- The comparative methods are not enough in Section 4.2. The authors may increase the conviction of the experiments by experimenting on more datasets.

- Difficulties in adjusting the weights of predicted image tokens and predicted language tokens in the training process, and the paper does not mention how the weights of image loss and text loss in the model are determined, but only mentions qualitatively that the weight of image loss should be greater. Besides, only two values for text loss weight is listed for comparison, and image loss is not mentioned in the context, which is not sufficient. Please clarify this.

**Summary Of The Paper:**

This paper proposed UPGen, a unified pre-trained model for both representation learning and generation, which is an encoder-only model based on masked Transformer for multimodal tasks. They show the potential of the masked token prediction model which can be directly used to generate images and language by iteratively re-masking and predicting masked tokens.

Following MaskGIT, they use VQGAN as the image tokenizer, which can tokenize image into tokens and generate image by discrete tokens. Then they combine sequence of image tokens and language tokens as input of ViT after several embedding layers. They use masked token prediction objective in both pre-training step and generation step.
And at the stage of iterative cross-modality generation, the author uses a confidence-based sampling mechanism following MaskGit. From experiment, the authors devote to demonstrating the usage of this model in serving as a representation learning model and generative model for image and language.

**Summary Of The Review:**

Although this work puts forward a simple and elegant way of generation, it needs more study of this one stage model to show the advantage. And the performance of this model is still far from SOTA, even though the author strengthen that they pre-train this model on a smaller dataset, they still need to show the comparison between the same dataset scale at least.

---

> ### Author Response · Authors · 2022-11-19
> **Author Response**
>
> We appreciate the reviewer’s helpful comments and suggestions. We address the concerns below.
>
> > This work does not show the advantage over other generative models. The quality of generated images do not perform SOTA results. Although the authors strengthen that they train the model on a small dataset, they still need to compare with the prior works on the same dataset scale to prove their model’s performance and transferability.
>
>
> We would like to emphasize that Upgen is capable of doing text-to-image, image-to-text generation, and learns representation, despite its simplicity.
> There are several important factors that contribute to the non-sota performance of Upgen.
> - Dataset. Our training dataset being considerably smaller and lower quality than prior works, for instance, Upgen is trained on 30M size public dataset while Parti is trained on 5B size private image-text datasets.
> - Pretrained model. Upgen is trained from scratch while prior works use pretrained models, for instance, Imagen uses a 11B sized pretrained T5 and Parti uses pretrained BERT.
> - Model size. Our largest Upgen model is 600M parameters, while Imagen has 13B, and Parti has 20B parameters.
>
> We believe that while Upgen is a non-sota model that is valuable to our research community, our source code will be made publicly available to facilitate further research on building simple and general purpose models from image and language data.
>
> Nonetheless, we fully agree with the reviewer that having better results would significantly strengthen our work.
> We have been working on scaling up our dataset from the current mixture of CC12M and Redcaps to larger scale LAION 400M dataset , since the training is very time consuming and compute intensive, we are not able to finish the training within the discussion window, but we will include the results in the next version.
>
>
> > Image loss weight is not mentioned.
>
>
> The image loss weight is 1 across the experiments. We have updated the revision to clarify, and we thank the reviewer for asking for the clarification.
>
>
> > The Linear Classification accuracy on ImageNet 1K does not achieve comparable performance to prior works.
>
>
> While we do not fully characterize the reason behind the observation, the hypothesis is that there is modality gap between image tokens and language tokens, we would like to emphasize that our model can do both cross-modal generation and representation, we hope the unified model could pave the way for future research in closing the modality gap on representation learning.

---

### Official Review · Reviewer_jhwP · 2022-10-22

**Confidence:** 4
**Correctness:** 2
**Technical Novelty And Significance:** 2
**Empirical Novelty And Significance:** 2
**Recommendation:** 3

**Clarity, Quality, Novelty And Reproducibility:**

The novelty is not significant. The paper is mostly clearly-written with sufficient implementation details.

**Strength And Weaknesses:**

Strengths:
- This paper is one of the first papers that combine masked image modeling with masked language modeling, which is an interesting direction.
- The proposed model shows the capability of image-to-text generation while also achieving reasonable representation learning performance.

Weaknesses:
1. The motivation - current text-to-image methods have drawbacks due to two-stage training - is not very convincing to me, for the following reasons.
    - The proposed method requires a pretrained VQGAN model, so technically it is also a two-stage method, similar as LDM or Parti.
    - The claimed "drawbacks" - extra hyperparameters and complexity - do not seem to be particularly harmful. On the contrary, a pre-trained image tokenizer or pre-trained text encoder might be necessary to ensure good image generation quality.

2. The experiments are insufficient to verify the advantage of UPGen over existing vision-language representation learning and text-to-image generation methods.
    - In Table 1, UPGen with image-text data achieves the same representation learning performance as UPGen with image-only data. This suggests that the additional language supervision is not effectively utilized. Thus UPGen offers no advantage compared to existing MIM methods such as BEIT.
    - In Table 1, the authors hypothesis that the superior performance of CLIP comes from the larger amount of training data. This hypothesis can be easily verified. The authors can pre-train CLIP on the same data as UPGen and make a fair comparison. The same goes for other baselines such as MAE.
    - More downstream vision-language tasks such as image-text retrieval and VQA are necessary to evaluate the vision-language representation learning performance.
    - The performance of UPGen is far worse than existing text-to-image generation methods. The authors attribute this to the smaller amount of pre-training data used. While it might be difficult to scale-up the pre-training data for UPGen, it is much easier to scale-down the pre-training data for existing methods and make a fair comparison.
    - The image captioning performance does not seem good. What is performance if the model is fine-tuned on image captioning datasets?

4. An important question has not been answered in the paper: whether the unification of representation learning and image generation brings benefit to each individual task?


**Summary Of The Paper:**

This paper combines two tasks - masked image modeling and masked language modeling - in a single framework, and trains a vision-language encoder to solve both tasks. The resulting model shows some capability for image-to-text generation, as well as reasonable representation learning performance.

**Summary Of The Review:**

This paper proposes an interesting idea similar to BEIT-3, which unifies MIM and MLM with a single model. Unfortunately, the experiments are not well-executed. The paper does not demonstrate the advantage of the proposed method over existing ones. Some claims are also not well-supported.

---

> ### Author Response · Authors · 2022-11-19
> **Author Response**
>
> We thank the reviewer for helpful comments. We address the concerns below.
>
>
> > In Table 1, UPGen with image-text data achieves the same representation learning performance as UPGen with image-only data. This suggests that the additional language supervision is not effectively utilized. Thus UPGen offers no advantage compared to existing MIM methods such as BEIT.
>
> While we do not fully characterize the reason behind the observation, we would like to emphasize that our model can do both cross-modal generation and representation. We hope the unified model can pave the way for future research in closing the modality gap on representation learning.
>
>
> > More downstream vision-language tasks such as image-text retrieval and VQA are necessary to evaluate the vision-language representation learning performance.
>
> We agree with the reviewer that these tasks are important to evaluate the capabilities of our method. However, in this paper, we focus mostly on the capability of unifying representation learning and generation in a single model with a simple training objective. We will add more evaluations in the next version of our paper.
>
>
> > The performance of UPGen is far worse than existing text-to-image generation methods. The authors attribute this to the smaller amount of pre-training data used. While it might be difficult to scale-up the pre-training data for UPGen, it is much easier to scale-down the pre-training data for existing methods and make a fair comparison.
>
> We thank the reviewer for the suggestion. Since the source code of Parti and Imagen is not publicly available, we were not able to reproduce their results. We have been working on scaling up our dataset from the current mixture of CC12M and Redcaps to LAION 300M dataset which is , since the training is very time consuming and compute intensive, we are not able to finish the training within the discussion window, but we will include the results in the next version.

---

### Official Review · Reviewer_s48X · 2022-10-24

**Confidence:** 3
**Correctness:** 3
**Technical Novelty And Significance:** 2
**Empirical Novelty And Significance:** Not applicable
**Recommendation:** 5

**Clarity, Quality, Novelty And Reproducibility:**

The paper is well-written and already in a good shape, but the technical novelty seems to be marginal. It does not provide code for reproduction.

**Details Of Ethics Concerns:**

Not applicable: The paper does not have any ethical considerations to address.

**Strength And Weaknesses:**

Strength
+ The paper is well-organized and easy to read.
+ The proposed method, UPGen, is the first to unifies pretraining for both representation learning and generation.

Weaknesses

It seems that the technical novelty of the proposed method is somewhat marginal: It throughly follows the training scheme of previous work, MaskGIT except that it uses additional modality, language. It does not have any contributions in terms of learning methods, model architectures and training/evaluation data. I believe that the model should competitive empirical results for publication in ICLR and the authors should provide the following things:
- I wonder why the linear classification result on ImageNet1K are worse than MAE using very similar objective even though the model is trained on much more image data than MAE. Do you have results using ImageNet1K as pretraining image data?
- I can't figure out the model's image-to-text ability just by looking at the results in Table 2. Could you provide apple-to-apple comparison results, in terms of training data or model capacity? You do not have to conduct new experiments. You can use numbers reported in previously published papers.
- The quality of generated outputs of image-to-text generation shown in Figure 6 is not good. Could you provide quantitative results for this task?
- You argue that the proposed method is the first framework that unifies pretraining for both representation learning and generation and you use large-scale paired or unpaired text data for pretraining. You need to provide empirical results on natural language processing tasks, not just image classification results on visual modality.

Minor comments:
- You should fix typos.
- Does the qualitative examples in Figure 6 come from CC12M or MS-COCO?

**Summary Of The Paper:**

The paper proposes a unified self-supervised learning framework for both representation learning and generation. The proposed approach first discretizes an input image into a sequence of tokens using VQGAN and concatenate them with text tokens. It then trains a bidirectional Transformer encoder using the masked token prediction objective. At inference, it generates image or text by iteratively re-masking and predicting the masked tokens for image or text generation tasks. The paper empirically validates the quality of learned visual representations on ImageNet1K classification and the model's generation ability on MS-COCO, both qualitatively and quantitatively.

**Summary Of The Review:**

I am not fully convinced that it is significant enough to warrant acceptance (see the weaknesses above). I will make the decision after seeing the rebuttal.

---

> ### Author Response · Authors · 2022-11-19
> **Author Response**
>
> We thank the reviewer for constructive feedback and helpful comments. We address the concerns below.
>
> > I wonder why the linear classification result on ImageNet1K are worse than MAE using very similar objective even though the model is trained on much more image data than MAE. Do you have results using ImageNet1K as pretraining image data?
>
> Because the model is encoder-only architecture, in contrast to encoder-decoder architecture in MAE where the extra decoder does the heavy lifting job of generation task, our model has smaller capacity for learning linearly separable  representations. We found that this linear classification gap can be easily eliminated by using a slighter larger encoder. We will include our results and this finding on ImageNet1K in the next revision.
>
>
> > I can't figure out the model's image-to-text ability just by looking at the results in Table 2. Could you provide apple-to-apple comparison results, in terms of training data or model capacity? You do not have to conduct new experiments. You can use numbers reported in previously published papers.
>
> The results in Table 2 are cited from the original papers. Since the source code and pre-trained models of prior works such as DALL-E 2, Parti and Imagen are not publically available, and many of them are trained on proprietary datasets, we were unable to reproduce their models and results.
>
> Upgen uses smaller public datasets and is significantly smaller.
> - Dataset. Our training dataset being considerably smaller and lower quality than prior works, for instance, Upgen is trained on 30M size public dataset while Parti is trained on 5B size private image-text datasets.
> - Pretrained model. Upgen is trained from scratch while prior works use pretrained models, for instance, Imagen uses a 11B sized pretrained T5 and Parti uses pretrained BERT.
> - Model size. Our largest Upgen model is 600M parameters, while Imagen has 13B, and Parti has 20B parameters.
>
> Despite these disadvantages in dataset and model size, Upgen achieves reasonable results across image-to-text, text-to-image, and representation learning.
>
> > Do the qualitative examples in Figure 6 come from CC12M or MS-COCO?
>
> The examples are randomly chosen from CC12M.

---

### Official Review · Reviewer_woHh · 2022-10-25

**Confidence:** 4
**Clarity, Quality, Novelty And Reproducibility:** Lack of novelty, and please revise ot…
**Correctness:** 3
**Technical Novelty And Significance:** 2
**Empirical Novelty And Significance:** 2
**Recommendation:** 3

**Strength And Weaknesses:**

Strength:

  1.This paper proposes a simple idea.
  2.This paper is well written and organized.

Weaknesses:

1.Lock of novelty. In contrast to DALL-E, this paper uses a BERT-like bidirectional encoding during the pretraining phase. And similar to BEiT-3, pretrain on both mono-modal and image-text pair data, and perform mask-and-reconstruct loss.
2.The reviewer thought the result in this paper was not very satisfactory.


**Summary Of The Paper:**

This paper presents a unified pre-trained model UPGen for both generation and representation learning.
Like DALL-E/BEiTv1-3, the image data are tokenized into dVAE/VQ-VAE tokens, and then is used to learn MLM on mono-modal data or image-text pair data with a random probability.


**Summary Of The Review:**

I prefer to reject this paper since the novelty, please check the weaknesses.

---

> ### Author Response · Authors · 2022-11-19
> **Author Response**
>
> We thank the reviewer for helpful feedback. We address the reviewer’s questions below.
>
> > Similar to BEiT-v3
>
> Upgen is an unified model for both representation learning and cross-modal generation (image generation, image-to-text, and text-to-image), while BEiT-v3 is a model only for representation learning.
> Due to the relative short gap between the ICLR deadline and the release of BEIT-v3 (a month), we believe that our work should be considered as concurrent to BEIT-v3.
>
>
> > The result in this paper was not very satisfactory
>
> We would appreciate the reviewer to point out which part of the results more specifically. We assume this is referring to the text-to-image generation results.
>
> Compared with state-of-the-art methods such as Parti and Imagen, we agree Upgen does not match their performance.
>
> We would like to emphasize that Upgen is capable of doing text-to-image, image-to-text generation, and learns representation, despite its simplicity.
>
> There are several important factors that contribute to the non-SOTA performance of Upgen.
> - Dataset: Our training dataset is considerably smaller and lower quality than prior work. For instance, Upgen is trained on a 30M size open-source dataset, while Parti is trained on a 5B size private image-text dataset that is not publicly available.
> - Pretrained model: Upgen is trained from scratch while prior works use pretrained models. For instance, Imagen uses a 11B sized pretrained T5 and Parti uses pretrained BERT.
> - Model size: Our largest Upgen model is 600M parameters, while Imagen has 13B, and Parti has 20B parameters.
>
>
>
> We believe that while Upgen does not achieve text-to-image performance, it is valuable to our research community as indicated by the reviewers. Our source code will be made publicly available to facilitate further research on building simple and general purpose models from image and language data.

---

### Author Response · Authors · 2022-11-19
**To All Reviewers**

First of all, we’d like to thank the reviewers for their constructive comments and feedback. We thank the reviewers for appreciating our results and method, and thinking that “simple idea” and “well written and organized” (reviewer woHh), “well organized and easy to read” and “Upgen is the first to unifies pretraining for both representation learning and generation” (reviewer s48X),  “an interesting direction” and “shows some capability for image-to-text generation, as well as reasonable representation learning performance” (reviewer jhwP), and “This way of text-prompt image generation and image-prompt text generation is simple but elegant” (reviewer DLC8).

The reviewers generally raise the concern that UPGen has worse performance compared to state-of-the-art text-to-image models (e.g. Parti, Imagen).
We would like to emphasize that Upgen is capable of doing text-to-image, image-to-text generation, and learns representation, despite its simplicity.
There are several important factors that contribute to the non-SOTA performance of Upgen.
- Dataset: Our training dataset is considerably smaller and lower quality than prior work. For instance, Upgen is trained on a 30M size open-source dataset, while Parti is trained on a 5B size private image-text dataset that is not publicly available.
- Pretrained model: UPGen is trained from scratch while prior works heavily rely on pretrained models. For instance, Imagen uses a 11B sized pretrained T5 and Parti uses pretrained BERT. We note that Upgen can easily leverage pretrained models by fine tuning a text encoder similar to Parti, which we leave as an interesting future work.
- Model size: Our largest Upgen model is 600M parameters, while Imagen has 13B, and Parti has 20B parameters.

We believe that while Upgen does not achieve state-of-the-art text-to-image performance, it is valuable to our research community as indicated by the reviewers. Our source code will be made publicly available to facilitate further research on building simple and general purpose models from image and language data.

Nonetheless, we fully agree with reviewers that having better results would strengthen our work.
We have been working on scaling up our dataset from the current mixture of CC12M and Redcaps to larger scale LAION 400M dataset. Since the training is very time consuming and computationally intensive, we are not able to finish the training within the discussion window, but we will include the results in the next version.

---

### Decision · Program_Chairs · 2023-01-20

**Decision:**

Reject

**Justification For Why Not Higher Score:**

1. The achievable performance is not satisfactory.
2. The results could not directly answer the main question: whether the unification of representation learning and image generation brings benefit to each individual task?

**Justification For Why Not Lower Score:**

1. The paper presents a unified framework based on a vision-language Transformer trained with a masked token prediction for representation learning and generation.
2. The proposed work first time investigates building a model for representation learning and generation., and obtains interesting results.

**Metareview: Summary, Strengths And Weaknesses:**

1. The paper presents a unified framework based on a vision-language Transformer trained with a masked token prediction for representation learning and generation.
2. The proposed work first time investigates building a model for representation learning and generation., and obtains interesting results.
3. The paper has some degree of scientific novelty, but may not meet ICLR's very high standards.
4. The achievable performance is not satisfactory.
5. The results could not directly answer the main question: whether the unification of representation learning and image generation brings benefit to each individual task?

**Summary Of Ac-Reviewer Meeting:**

The scores from the reviewers are quite consistent. Although the authors have replied to several comments, the reviewers still think the paper could not meet the very high standard of ICLR. Therefore, there was no AC-reviewer meeting for this paper.